# Dementia and the risk of short-term readmission and mortality after a pneumonia admission

Susanne Boel Graversen[1,2]*, Henrik Schou Pedersen[1], Annelli Sandbaek[2,3], Catherine Hauerslev Foss[4], Victoria Jane Palmer[5], Anette Riisgaard Ribe[1]

1 Research Unit for General Practice, Aarhus, Denmark, 2 Department of Public Health, Aarhus University, Aarhus, Denmark, 3 Steno Diabetes Center Aarhus, Aarhus, Denmark, 4 Department of Geriatrics, Aarhus University Hospital, Aarhus, Denmark, 5 The Department of General Practice, Melbourne Medical School, University of Melbourne, Melbourne, Australia

* sugrav@ph.au.dk

**Data Availability Statement:** Due to restrictions related to Danish law and protecting patient privacy, the combined set of data as used in this study can only be made available through a trusted

## Abstract

### Background

At time of discharge after a pneumonia admission, care planning for older persons with dementia is essential. However, care planning is limited by lack of knowledge on the short-term prognosis.

### Aim

To investigate 30-day mortality and readmission after hospital discharge for pneumonia in persons *with* versus *without* dementia, and to investigate how these associations vary with age, time since discharge, and medication use.

### Methods

Using the Danish registries, we investigated 30-day mortality and readmission in persons (+65 years) discharged after pneumonia in 2000–2016 (N = 298,872). Adjusted mortality rate ratios (aMRRs) and incidence rate ratios (aIRRs) were calculated for persons with versus without dementia, and we investigated if these associations varied with use of benzodiazepines, opioids, and antipsychotics, and with age and time since discharge.

### Results

Among 25,948 persons with dementia, 4,524 died and 5,694 were readmitted within 30 days. The risk of 30-day mortality was 129% higher (95% CI 2.21–2.37) in persons with versus without dementia after adjustment for sociodemographic characteristics, admission-related factors, and comorbidities. Further, the highest mortality risk was found in persons with both dementia and use of antipsychotics (aMRR: 3.39, 95% CI 3.19–3.59); 16% of deaths in this group could not be explained by the independent effect of each exposure. In those with dementia, the highest aMRRs were found for the youngest and for the first days after discharge. The risk of 30-day readmission was 7% higher (95% CI 1.04–1.10) in

third party, Statistics Denmark. This state organisation holds the data used for this study. University-based Danish scientific organisations can be authorized to work with data within Statistics Denmark and such organisation can provide access to individual scientists inside and outside of Denmark. Requests for data may be sent to Statistics Denmark: http://www.dst.dk/en/OmDS/organisation/TelefonbogOrg.aspx?kontor=13&tlfbogsort=sektion or the Danish Data Protection Agency: https://www.datatilsynet.dk/kontakt/.

**Funding:** SG was funded by unrestricted grants from the General Practice Research Foundation of the Central Denmark Region (Region Midtjyllands Praksisforskningsfond), the Danish General Practice Fund (Fonden for Almen Praksis), Public Health in Central Denmark Region - a collaboration between municipalities and the Central Denmark Region (Folkesundhed i Midten), the (Danish) Health Foundation (Helsefonden), the Danish Foundation for Physicians (Lægefonden) under the A.P. Moeller Foundations, the Beckett Foundation (Beckett-fonden), the Knud Hoejgaards Fond, the Graduate School of Health Aarhus University, and the Korning Foundation. ARR was funded by an unrestricted grant from the Novo Nordisk Foundation. The funders had no role in the study design, data collection, data analysis, data interpretation, or writing of the report.

**Competing interests:** The authors have declared that no competing interests exist.

persons with versus without dementia. In those with dementia, the highest aIRRs were found for the first days after discharge.

## Conclusions

Dementia was associated with higher short-term mortality after pneumonia, especially in users of antipsychotics, and with slightly higher readmission, especially in the first days after discharge. This is essential knowledge in the care planning for persons with dementia who are discharged after a pneumonia admission.

## Introduction

Persons with dementia are at heightened risk of being hospitalized for pneumonia. This may be explained by dementia-related characteristics (e.g. dysphagia and impaired functional status) and more frequent use of sedative medications, which are all established risk factors for developing pneumonia [1–3]. Around one in eight persons with dementia dies during their pneumonia admission [4, 5], but the prognosis for those discharged from hospital remains underexplored.

Further, persons with dementia are often prescribed psycholeptics, such as benzodiazepines and antipsychotics, due to behavioral symptoms (e.g. agitation and aggression), which are more frequent in advanced dementia [6, 7]. Pain is recognized as an important cause of agitation in persons with dementia, [8] and a substantial increase in the use of opioids among older persons with dementia has been observed in Denmark during the past decades [9]. Use of psycholeptics and opioids may cause respiratory harm due to the sedative effects [10–12], which could possibly contribute to poorer outcomes after a pneumonia admission [13].

Nevertheless, no studies have investigated the association between dementia and both short-term mortality and readmission after discharge for a pneumonia admission. Knowledge is urgently needed on the prognosis of patients with dementia who transfer from secondary to primary care management. Such knowledge could inform shared decision-making between healthcare professionals, patient and caregivers and enhance care planning at discharge.

We aimed to investigate the risk of 30-day mortality and 30-day readmission after a pneumonia admission in persons with dementia versus persons without. We examined whether these associations varied with age and time since discharge in subgroups defined by users and non-users of benzodiazepines, opioids, and antipsychotics.

## Methods

### Study design and setting

We conducted a population-based cohort study on all registered pneumonia admissions among older adults (65–99 years) in Denmark from January 1, 2000 to December 1, 2016, using information from the nationwide Danish registries. In Denmark, each citizen is provided with a unique identification number, which allows for accurate linkage of all prospectively collected health-care data in these registries. Denmark provides tax-funded universal health care for all citizens with general practitioners in primary care acting as gatekeepers to specialized care [14]. The hospital sector is predominantly public, and private hospitals account for less than 1% of hospital beds [15]. A co-funding model was introduced in 2007 for hospital treatment of citizens to enhance preventive measures and health promotion in the municipalities. The municipalities provide free-of-charge home care and home nursing facilities to citizens requiring this service [15]. For citizens with an extensive need of care due to

functional or cognitive decline, the municipalities provide permanent residency at nursing homes, which are staffed by health care professionals with 24/7 care available. [15]

This study was approved by the Danish Data Protection Agency and the Danish Health Data Authority. Ethics approval and informed consent of individuals were not required since data were analyzed anonymously and based on de-identified register-data.

## Study population

The study population consisted of all Danish citizens aged 65–99 years, who were discharged after a hospital admission (index admission) with a primary or secondary diagnosis of pneumonia as defined by the allocation of diagnostic codes (ICD-10: J12-J18, A709, or A481 [16]). In order to include an older, geriatric population (versus first-time pneumonia admissions only), reentry to the cohort was allowed. Yet, in order to study incident pneumonia, twelve months after end of follow-up (i.e. 30 days after discharge) had to pass before a person could re-enter the cohort with a new pneumonia admission. To ensure complete medical history, a person was excluded if the person had not resided in Denmark for at least the five years preceding their index admission.

## Data sources

We obtained data on sex, age, cohabitation, migration, and death from the Danish Civil Registration System [17], data on education, nursing home, and home care from Statistics Denmark [18], data on index admission and readmissions from the National Patient Registry [19], data on comorbidities from the National Patient Registry, the Danish Psychiatric Central Research Register [20], and the Danish National Prescription Registry [21], and data on selected medications of clinical interest for the investigated associations from the Danish National Prescription Registry. All registries have been described in detail elsewhere [15].

## Outcome

Outcomes of interest were death within 30 days after discharge (30-day mortality) and readmission within 30 days after discharge (30-day readmission).

## Exposure

We defined dementia as a diagnosis of dementia (ICD-8: 290.09–11, 290.18–19, 293.09–19, ICD-10: F00.0–00.9, G30.0–30.9, F01.0–01.9, F02.0, F03.9, G31.9, G31.8) and/or a redeemed prescription for an anti-dementia drug (i.e. acetyl cholinesterase inhibitor drugs and memantine; ATC: N06DA01-04, N06DA52, N06DX01) at any given time prior to the index admission [22, 23].

## Covariates

We assessed socioeconomic factors, type of residency, extent of home care, comorbidities (according to a modified version of a previously developed comorbidity index [24]) and selected medications on day of admission (S1–S4 Appendices), and factors related to the index admission (i.e. length of stay and type of pneumonia diagnosis) on day of discharge (S5 Appendix) (Table 1). Data on educational level, type of residency, and extent of home care was available only for 2012–2016. Apart from medication use as specified in the comorbidity index [24], other medication use, including benzodiazepines, opioids, and antipsychotics, was defined as at least one prescription redemption within the six months preceding the index admission (S4 Appendix).

**Table 1. Study cohort characteristics stratified on dementia status in 298,872 admissions.**

| | Without dementia No (%) | With dementia No (%) | Total No (%) |
|---|---|---|---|
| **Total number of admissions** | 272,924 (100.0) | 25,948 (100.0) | 298,872[a] (100.0) |
| **Socioeconomic factors** | | | |
| Sex, male | 136,172 (49.9) | 12,261 (47.3) | 148,433 (49.7) |
| Age (years) | | | |
| 65–74 | 94,063 (34.5) | 4,041 (15.6) | 98,104 (32.8) |
| 75–84 | 110,584 (40.5) | 11,470 (44.2) | 122,054 (40.8) |
| 85–99 | 68,277 (25.0) | 10,437 (40.2) | 78,714 (26.3) |
| Calendar period | | | |
| 2000–2003 | 53,310 (19.5) | 3,538 (13.6) | 56,848 (19.0) |
| 2004–2007 | 58,849 (21.6) | 5,405 (20.8) | 64,254 (21.5) |
| 2008–2011 | 64,146 (23.5) | 6,588 (25.4) | 70,734 (23.7) |
| 2012–2016 | 96,619 (35.4) | 10,417 (40.1) | 107,036 (35.8) |
| Cohabitation, living alone | 145,334 (53.3) | 16,010 (61.7) | 161,344 (54.0) |
| Education[b] (years) | | | |
| > 15 | 10,054 (10.4) | 1,020 (9.8) | 11,074 (10.3) |
| 10–15 | 33,797 (35.0) | 3,404 (32.7) | 37,201 (34.8) |
| < 10 | 46,681 (48.3) | 5,205 (50.0) | 51,886 (48.5) |
| Unknown | 6,079 (6.3) | 788 (7.6) | 6,867 (6.4) |
| **Type of residency and home care** | | | |
| Type of residency, nursing home[c] | 7,780 (8.1) | 4,159 (39.9) | 11,939 (11.2) |
| Home care[d], personal care, dependent | 19,058 (19.7) | 2,825 (27.1) | 21,883 (20.4) |
| Home care[d], practical help, dependent | 22,526 (23.3) | 2,213 (21.2) | 24,739 (23.1) |
| **Index admission** | | | |
| Length of stay (days) | | | |
| 0–2 | 37,035 (13.6) | 4,519 (17.4) | 41,554 (13.9) |
| 3–7 | 102,732 (37.6) | 10,346 (39.9) | 113,078 (37.8) |
| > 7 | 133,157 (48.8) | 11,083 (42.7) | 144,240 (48.3) |
| Pneumonia diagnosis, secondary to another primary diagnosis | 86,018 (31.5) | 7,265 (28.0) | 93,283 (31.2) |
| **Comorbidities[e]** | | | |
| Non-psychiatric comorbidities | | | |
| Hypertension | 147,002 (53.9) | 14,624 (56.4) | 161,626 (54.1) |
| Dyslipidemia | 32,279 (11.8) | 3,098 (11.9) | 35,377 (11.8) |
| Ischemic heart disease | 77,451 (28.4) | 6,889 (26.5) | 84,340 (28.2) |
| Atrial fibrillation | 52,272 (19.2) | 5,244 (20.2) | 57,516 (19.2) |
| Heart failure | 43,721 (16.0) | 3,840 (14.8) | 47,561 (15.9) |
| Peripheral artery occlusive diseases | 33,552 (12.3) | 2,841 (10.9) | 36,393 (12.2) |
| Stroke | 45,540 (16.7) | 8,053 (31.0) | 53,593 (17.9) |
| Diabetes mellitus | 43,602 (16.0) | 4,292 (16.5) | 47,894 (16.0) |
| Chronic pulmonary disease | 83,059 (30.4) | 4,780 (18.4) | 87,839 (29.4) |
| Ulcer/chronic gastritis | 25,553 (9.4) | 2,959 (11.4) | 28,512 (9.5) |
| Dysphagia | 5,093 (1.9) | 623 (2.4) | 5,716 (1.9) |
| Chronic liver disease | 3,873 (1.4) | 397 (1.5) | 4,270 (1.4) |
| Inflammatory bowel disease | 4,147 (1.5) | 395 (1.5) | 4,542 (1.5) |
| Diverticular disease of the intestine | 18,825 (6.9) | 1,860 (7.2) | 20,685 (6.9) |
| Chronic kidney disease | 13,527 (5.0) | 1,087 (4.2) | 14,614 (4.9) |
| Prostate disorders | 37,577 (13.8) | 4,216 (16.2) | 41,793 (14.0) |
| Connective tissue disorder | 19,962 (7.3) | 1,656 (6.4) | 21,618 (7.2) |

*(Continued)*

**Table 1.** (Continued)

| | Without dementia No (%) | With dementia No (%) | Total No (%) |
|---|---|---|---|
| Osteoporosis | 39,673 (14.5) | 4,239 (16.3) | 43,912 (14.7) |
| Painful condition | 78,870 (28.9) | 10,145 (39.1) | 89,015 (29.8) |
| Anemias | 17,882 (6.6) | 2,119 (8.2) | 20,001 (6.7) |
| Cancer | 43,518 (15.9) | 2,181 (8.4) | 45,699 (15.3) |
| Multiple sclerosis | 1,194 (0.4) | 85 (0.3) | 1,279 (0.4) |
| Parkinson's disease | 4,323 (1.6) | 1,733 (6.7) | 6,056 (2.0) |
| Epilepsy | 5,367 (2.0) | 1,273 (4.9) | 6,640 (2.2) |
| Neuropathies | 4,862 (1.8) | 323 (1.2) | 5,185 (1.7) |
| Psychiatric comorbidities | | | |
| Mood, stress or anxiety-related disorder | 9,409 (3.4) | 2,514 (9.7) | 11,923 (4.0) |
| Psychological distress | 47,079 (17.2) | 341 (1.3) | 47,420 (15.9) |
| Alcohol problems | 3,448 (1.3) | 632 (2.4) | 4,080 (1.4) |
| Substance abuse | 502 (0.2) | 87 (0.3) | 589 (0.2) |
| Bipolar affective disorder | 2,835 (1.0) | 709 (2.7) | 3,544 (1.2) |
| Schizophrenia/schizo-affective disorder | 1,720 (0.6) | 329 (1.3) | 2,049 (0.7) |
| Comorbidities (number) | | | |
| 0 | 18,362 (6.7) | 1,301 (5.0) | 19,663 (6.6) |
| 1–2 | 82,220 (30.1) | 7,534 (29.0) | 89,754 (30.0) |
| 3–4 | 91,856 (33.7) | 9,073 (35.0) | 100,929 (33.8) |
| > 4 | 80,486 (29.5) | 8,040 (31.0) | 88,526 (29.6) |
| **Medication use**[f] | | | |
| Antibiotics | 148,746 (54.4) | 15,360 (59.2) | 164,106 (54.9) |
| Antithrombotic drugs | 134,953 (49.4) | 14,745 (56.8) | 149,698 (50.1) |
| Glucocorticoids | 51,560 (18.9) | 2,484 (9.6) | 54,044 (18.1) |
| Antineoplastic agents and immunosuppressants | 5,019 (1.8) | 228 (0.9) | 5,247 (1.8) |
| Benzodiazepines | 82,204 (30.1) | 6,872 (26.5) | 89,076 (29.8) |
| Opioids | 79,076 (29.0) | 7,553 (29.1) | 86,629 (29.0) |
| Antipsychotics | 15,599 (5.7) | 6,246 (24.1) | 21,845 (7.3) |

All variables shown in column percentages.

[a]298,872 pneumonia admissions in 246,498 individuals.

[b]Highest level of education was categorized into three levels (S1 Appendix), available only in 2012–2016 (N = 106,949) and assessed on the date of index admission.

[c]Type of residency was dichotomized into nursing home yes/no (S2 Appendix), available only in 2012–2016 (N = 106,949) and assessed in the month preceding index admission.

[d]Home care was dichotomized into personal care or practical help (S2 Appendix), available only in 2012–2016 (N = 106,949) and assessed in the month preceding index admission.

[e]Comorbidity was categorized according to a modified version of a previously developed algorithm[24] (S3 and S4 Appendices), available in the whole study period and assessed on the date of index admission.

[f]Medication use was dichotomized, available in the whole study period and assessed as at least one redeemed prescription in the six months preceding index admission.

## Statistical analysis

Follow-up started one day after the date of discharge. This approach was taken to avoid classifying hospital transfers as readmissions and thereby overestimating the number of events. Individuals contributed with at-risk time until date of death, date of readmission (when analyzing readmission only), date of emigration, or end of 30-day follow-up, whichever came first.

We used a log-linear Poisson model to estimate mortality rate ratios (MRRs) and incidence rate ratios (IRRs) with the logarithm of person days as the offset. We applied cluster robust

variance estimation to account for the fact that persons could be included in the dataset more than once [25]. In five a priori chosen nested models, the MRRs and IRRs were successively adjusted for: 1) sex, age, calendar period, cohabitation status, length of stay, type of pneumonia diagnosis, and time since discharge, 2) also non-psychiatric comorbidities, 3) also psychiatric comorbidities and alcohol and substance abuse (a priori main model), 4) also use of antibiotics, antithrombotic drugs, glucocorticoids, antineoplastic agents, and immunosuppressants, and 5) also use of benzodiazepines, opioids, and antipsychotics.

First, we estimated the overall adjusted MRRs (aMRRs) for 30-day mortality and adjusted IRRs (aIRRs) for 30-day readmission comparing persons with and without dementia. Second, we estimated the mortality rates, incidence rates, aMRRs, and aIRRs for 30-day mortality and 30-day readmission for each of the selected medications (benzodiazepines, opioids, and antipsychotics) across four subgroups (i.e. combinations of without/with dementia and without/with medication use), both overall and as a function of age and days since discharge. To allow for non-linearity, we used restricted cubic splines [26]. In the overall analysis, we further estimated the attributable proportion (AP) due to interaction among the doubly exposed (i.e. individuals with dementia and with use of the given medication) as a measure of the excess aMRR and aIRR, which was not explained by the independent magnitude of either exposure [27]. Moreover, we calculated the excess number of events among the doubly exposed caused by interaction by multiplying the number of events with the estimated AP.

In a supplementary analysis, we calculated the aMRRs and aIRRs for 30-day mortality and readmission comparing persons with and without dementia in a restricted study period (2012 onwards) to further adjust our main model for educational level, type of residency, and extent of home care.

Finally, in sensitivity analyses, we calculated the aMRRs and aIRRs for 30-day mortality and readmission comparing persons with and without dementia using alternative definitions of dementia and of current use of benzodiazepines, opioids, and antipsychotics. First, we restricted the definition of dementia in five separate analyses to include only: 1) those identified by diagnoses (i.e. hospital contact), 2) those identified by use of anti-dementia drugs, 3) those identified by both a diagnosis and use of anti-dementia drugs, 4) those diagnosed above the age of 60 years [28], and 5) those diagnosed within five years prior to the index admission. Second, we restricted the definition of use of benzodiazepines, opioids, and antipsychotics to include at least one redeemed prescription within a restricted time frame of four months prior to index admission.

All analyses were performed with Stata 15 (Stata Corporation, College Station, TX).

## Results

The study population included 298,872 pneumonia admissions in 246,498 individuals, hereof 25,948 admissions (8.7%) with dementia (Table 1). Around one fourth of those with and of those without dementia used benzodiazepines and opioids, whereas the proportion of antipsychotic users was higher in those with dementia versus those without (24.1% vs 5.7%).

### 30-day mortality

In total, 17.4% (N = 4,524) of patients with dementia died within 30 days after discharge for pneumonia compared with 7.3% (N = 19,821) of those without dementia. After adjustments, this corresponded to an overall aMRR of 2.29 (95% CI 2.21–2.37) in our main model (Table 2). This association was persistent throughout all five models (Table 2) and throughout sensitivity analyses with various definitions of dementia (S1 Table).

**Table 2. Stepwise adjusted mortality rate ratios (aMRRs) for the risk of 30-day mortality and stepwise adjusted incidence rate ratios (aIRRs) for the risk of 30-day readmission in pneumonia patients with dementia (N = 25,948) versus those without (N = 272,924) in 298,872 admissions.**

| | 30-day mortality | | | | | |
|---|---|---|---|---|---|---|
| | Deaths, N (%) | Model 1[a] aMRR (95% CI) | Model 2[b] aMRR (95% CI) | Model 3[c] aMRR (95% CI) | Model 4[d] aMRR (95% CI) | Model 5[e] aMRR (95% CI) |
| Without dementia | 19,821 (7.3) | 1 | 1 | 1 | 1 | 1 |
| With dementia | 4,524 (17.4) | 2.30 (2.22; 2.37) | 2.21 (2.13; 2.29) | **2.29 (2.21; 2.37)** | 2.31 (2.22; 2.39) | 2.13 (2.06; 2.21) |
| | 30-day readmission | | | | | |
| | Readmissions, N (%) | Model 1[a] | Model 2[b] | Model 3[c] | Model 4[d] | Model 5[e] |
| | | aIRR (95% CI) | aIRR (95% CI) | aIRR (95% CI) | aIRR (95% CI) | aIRR (95% CI) |
| Without dementia | 62,954 (23.1) | 1 | 1 | 1 | 1 | 1 |
| With dementia | 5,694 (21.9) | 1.05 (1.02; 1.08) | 1.07 (1.05; 1.11) | **1.07 (1.04; 1.10)** | 1.08 (1.05; 1.11) | 1.07 (1.04; 1.10) |

Abbreviations: MRR: mortality rate ratio; IRR: incidence rate ratio; CI: confidence interval.

[a]Adjusted for sex, age, calendar period, cohabitation status, length of stay, type of pneumonia diagnosis, and time since discharge.

[b]Further adjusted for somatic comorbidities.

[c]Further adjusted for psychiatric comorbidities and alcohol/substance abuse. Bold values = main model.

[d]Further adjusted for use of antibiotics, antithrombotic drugs, glucocorticoids, antineoplastic agents, and immunosuppressants in the previous six months (available data on medications not already adjusted for in the comorbidity index).

[e]Further adjusted for use of benzodiazepines, opioids, and antipsychotics in the previous six months.

## 30-day mortality among users of benzodiazepines, opioids, or antipsychotics

For each of the selected medications (benzodiazepines, opioids, and antipsychotics), we found higher aMRRs for those with use of the medication, even higher for those with dementia, and the highest aMRRs for those with both dementia and use of the medication (benzodiazepines: aMRR 2.56, 95% CI 2.40–2.72; opioids: aMRR 2.95, 95% CI 2.78–3.13; antipsychotics: aMRR: 3.39, 95% CI 3.19–3.59) compared with those without dementia and no medication use. For those with both dementia and use of antipsychotics, the AP was 0.16 (95% CI 0.10–0.22), implying that 16% of deaths (N = 220) in this group could not be explained by the independent magnitude of each of the two exposures (Table 3). No significant interaction was found between dementia and benzodiazepines and opioids, respectively. These results were persistent when applying a restrictive definition of medication use (S2 Table).

Across all exposures of dementia and medication use, the mortality rate increased with age and decreased with number of days since discharge. In those with dementia, the highest aMRRs were found for the youngest in the cohort and for the first days after discharge, regardless of the medication under study. For all ages and time since discharge, those with both dementia and use of benzodiazepines or opioids tended to have slightly higher aMRRs than those with dementia only, whereas those with both dementia and use of antipsychotics had substantially higher aMRRs than those with dementia only (Figs 1 and 2).

## 30-day readmission

In total, 21.9% (N = 5,694) of patients with dementia were readmitted within 30 days after discharge compared with 23.1% (N = 62,954) of those without dementia. After adjustments, this corresponded to an overall aIRR of 1.07 (95% CI 1.04–1.10) in our main model (Table 2). This association was persistent throughout all five models (Table 2) and throughout sensitivity analyses with various definitions of dementia (S3 Table).

**Table 3. Adjusted mortality rate ratios (aMRRs) for the risk of 30-day mortality and adjusted incidence rate ratios (aIRRs) for the risk of 30-day readmission in pneumonia patients with dementia (without medication use), with use of benzodiazepines, opioids or antipsychotics (without dementia), or with both, compared with those with neither exposure in 298,872 admissions.**

| | **30-day mortality** | | | | | |
|---|---|---|---|---|---|---|
| | **Dementia** | | | | **Attributable proportion due to interaction[b] (95% CI)** | **Excess number of events due to interaction[c] (95% CI)** |
| | **No** | | **Yes** | | | |
| | **Medication** | | **Medication** | | | |
| | **No** | **Yes** | **No** | **Yes** | | |
| | aMRR[a] (95% CI) | aMRR[a] (95% CI) | aMRR[a] (95% CI) | aMRR[a] (95% CI) | | |
| Benzodia-zepines | 1 | 1.21 (1.17; 1.25) | 2.38 (2.28; 2.48) | 2.56 (2.40; 2.72) | -0.01 (-0.08; 0.06) | -14 (-97; 70) |
| Opioids | 1 | 1.53 (1.48; 1.58) | 2.54 (2.43; 2.64) | 2.95 (2.78; 3.13) | -0.04 (-0.10; 0.03) | -57 (-153; 40) |
| Antipsy-chotics | 1 | 1.72 (1.63; 1.81) | 2.12 (2.04; 2.21) | 3.39 (3.19; 3.59) | 0.16 (0.10; 0.22) | 220 (142; 298) |
| | **30-day readmission** | | | | | |
| | **Dementia** | | | | **Attributable proportion due to interaction[b] (95% CI)** | **Excess number of events due to interaction[c] (95% CI)** |
| | **No** | | **Yes** | | | |
| | **Medication** | | **Medication** | | | |
| | **No** | **Yes** | **No** | **Yes** | | |
| | aIRR[a] (95% CI) | aIRR[a] (95% CI) | aIRR[a] (95% CI) | aIRR[a] (95% CI) | | |
| Benzodia-zepines | 1 | 1.12 (1.10; 1.14) | 1.09 (1.05; 1.13) | 1.15 (1.09; 1.21) | -0.05 (-0.11; 0.02) | -72 (-170; 26) |
| Opioids | 1 | 1.22 (1.20; 1.24) | 1.12 (1.08; 1.16) | 1.19 (1.13; 1.25) | -0.12 (-0.18; -0.06) | -217 (-328; -106) |
| Antipsy-chotics | 1 | 1.10 (1.06; 1.14) | 1.07 (1.04 1.11) | 1.10 (1.03; 1.16) | -0.07 (-0.15; 0.00) | -97 (-194; -1) |

Abbreviations: aMRR: adjusted mortality rate ratio; aIRR: adjusted incidence rate ratio; CI: Confidence interval, AP: attributable proportion.

[a] Adjusted for sex, age, calendar period, cohabitation status, length of stay, type of pneumonia diagnosis, time since discharge, somatic comorbidities, psychiatric comorbidities, and alcohol/substance abuse.

[b] The attributable proportion was calculated as (example shown for aMRR):

$AP = (aMRR_{(dementia+\ medication+)} - aMRR_{(dementia+\ medication-)} - aMRR_{(dementia-\ medication+)} + 1) / aMRR_{(dementia+\ medication+)}.$

[c] The excess number of events due to interaction was calculated as $N_{excess} = $ Number of events$_{(dementia+\ medication+)}$ * AP.

## 30-day readmission among users of benzodiazepines, opioids, or antipsychotics

For each of the selected medications, we found that those using medications, those with dementia, and those with both dementia and medication use, all had higher aIRRs, ranging from 1.07 to 1.22, compared with those without dementia and no medication use. A negative interaction was seen for those with both dementia and use of opioids (AP: -0.12, 95% CI: -0.18; -0.06), implying that the effect of combining the two exposures was lower than expected based on the summation of the independent effects of each exposure (Table 3). No significant interaction was found between dementia and use of benzodiazepines or antipsychotics, respectively. These results were persistent when we applied a restrictive definition of medication use (S4 Table).

Across all exposures of dementia and medication use, the readmission rate decreased with age and number of days since discharge. In those with dementia, the highest aIRRs were found for the first days after discharge and tended to be found for the youngest in the cohort,

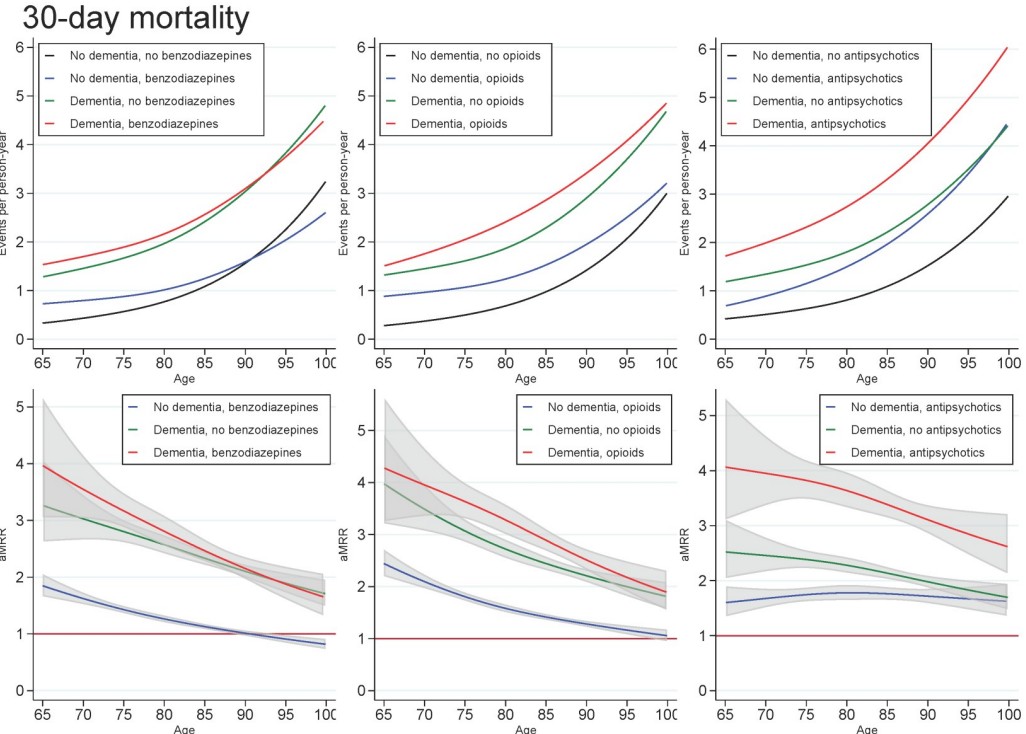

**Fig 1. Unadjusted mortality rates and adjusted mortality rate ratios for patients with dementia (without medication use), patients with use of benzodiazepines, opioids or antipsychotics (without dementia), and patients with both or neither exposures, plotted as a function of age.** Abbreviations: aMRR: adjusted mortality rate ratios. MRRs adjusted for sex, age, calendar period, cohabitation status, length of stay, type of pneumonia diagnosis, time since discharge, somatic comorbidities, psychiatric comorbidities, and alcohol/substance abuse.

regardless of the medication under study. For all ages and time since discharge, those with both dementia and use of medication tended to have similar aIRRs as those with dementia only (Figs 3 and 4).

## 30-day mortality and readmission in 2012–2016

In a supplementary analysis, in which the study period was restricted to 2012 onwards, our results were essentially unchanged after additional adjustments for educational level, extent of home care, and type of residency (S5 and S6 Tables).

## Discussion

In this large cohort of older patients discharged with pneumonia in Denmark, we found that the short-term mortality was 129% higher in those with dementia versus those without. The highest mortality risks were seen in individuals with both dementia and use of benzodiazepines, opioids, or antipsychotics. Among those with both dementia and use of antipsychotics, 16% of deaths could not be explained by the independent magnitude of each exposure. In those with dementia, the highest mortality risks were seen in the youngest of the cohort and for the first days after discharge, regardless of the medication under study. Further, we found that the short-term readmission was 7% higher for those with dementia versus those without. In those with dementia, the highest readmission risks were found for the first days after discharge, regardless of the medication under study.

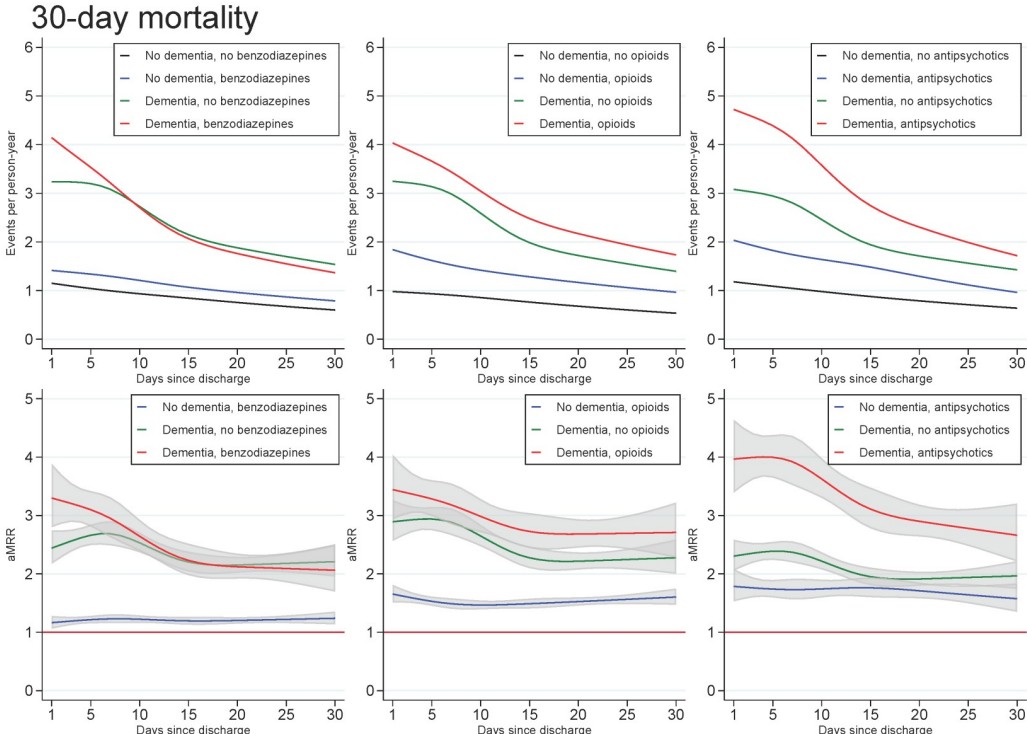

**Fig 2. Unadjusted mortality rates and adjusted mortality rate ratios for patients with dementia, (without medication use), patients with use of benzodiazepines, opioids or antipsychotics (without dementia), and patients with both or neither exposures, plotted as a function of days since discharge.** Abbreviations: aMRR: adjusted mortality rate ratios. MRRs adjusted for sex, age, calendar period, cohabitation status, length of stay, type of pneumonia diagnosis, time since discharge, somatic comorbidities, psychiatric comorbidities, and alcohol/substance abuse.

There may be several reasons why patients with dementia who have survived their pneumonia admission have a higher risk of 30-day mortality compared with those without dementia. First, during an acute hospitalization, persons with dementia are more prone to experience functional decline [29], delirium [30], falls [31], and dysphagia, [32] which are all predictors of mortality in older adults [33–36]. Second, persons with dementia often experience increased cognitive decline during admission [37], which can further impair their ability to express symptoms of disease and reach out for help, thereby increasing the risk of inadequate diagnostics and treatments in the post-discharge period. Third, among those with dementia, users of benzodiazepines, opioids, and antipsychotics had overall higher aMRRs than non-users. These medications are often used for managing behavioral symptoms in dementia [38–40], which are frequently present in more advanced stages of the disease [7]. Users of these medications may therefore comprise a more frail group of patients with dementia, which could explain some of the increased mortality among users of these medications in this study. However, benzodiazepines, opioids, and antipsychotics can cause sedation, dysphagia (which can lead to aspiration), and respiratory depression [10, 41–43], and therefore, we cannot rule out that some of the increased mortality observed in users of these medications, especially antipsychotics, could be due to drug-related adverse effects. Likewise, there may be several reasons why patients with dementia have a slightly higher risk of 30-day readmission compared with those without dementia. First, we found that the dementia-associated aIRRs were highest in the immediate days after discharge, which could indicate that in-hospital factors, such as acute illness burden and inpatient care process factors, are not sufficiently targeted in persons with

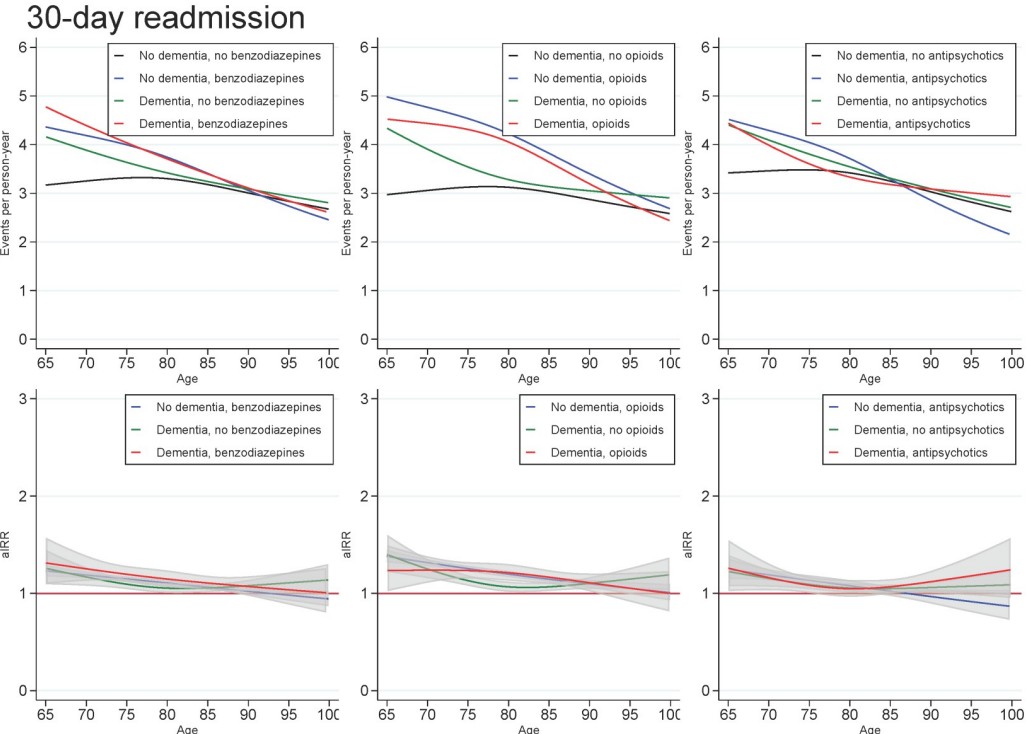

**Fig 3. Unadjusted readmission rates and adjusted incidence (readmission) rate ratios for patients with dementia (without medication use), patients with use of benzodiazepines, opioids or antipsychotics (without dementia), and patients with both or neither exposures, plotted as a function of age.** Abbreviations: aIRR: adjusted incidence (readmission) rate ratios. IRRs adjusted for sex, age, calendar period, cohabitation status, length of stay, type of pneumonia diagnosis, time since discharge, somatic comorbidities, psychiatric comorbidities, and alcohol/substance abuse.

dementia before discharge [44]. As dementia is an illness with an established evidence base for reduced life expectancy [45], very early readmissions could also indicate that attention has not been paid to setting goals of care and planning ahead (e.g. advanced care planning in the most severely affected individuals with dementia). Second, the risk of readmission tended to be highest among the youngest in the cohort because the eldest and most frail persons in Denmark are often attempted to treat at home (including nursing homes) to avoid burdensome transitions in and out of hospitals [46]. Use of benzodiazepines, opioids or antipsychotics did not markedly impact these associations, indicating that use of these medications is not feasible as a marker of readmission risk in this patient group.

This study is the first to investigate the dementia-associated mortality within 30 days after discharge for a pneumonia admission, thereby providing knowledge on the short-term prognosis in persons with dementia in the transition from hospital to home and primary care management. Although previous studies have found dementia-associated 30-day readmission proportions similar to ours (23–25% vs 22%) [47, 48], only one study has analyzed the 30-day readmission risk and found a three times higher risk in those with dementia versus those without [48]. However, as this study included only individuals with a primary diagnosis of pneumonia in a predominately insurance-based health-care system, this risk estimate is hardly comparable to that of our study.

Our study had several strengths, including the large population-based cohort and virtually no loss to follow-up, which practically eliminated the risk of selection bias. Our key variables

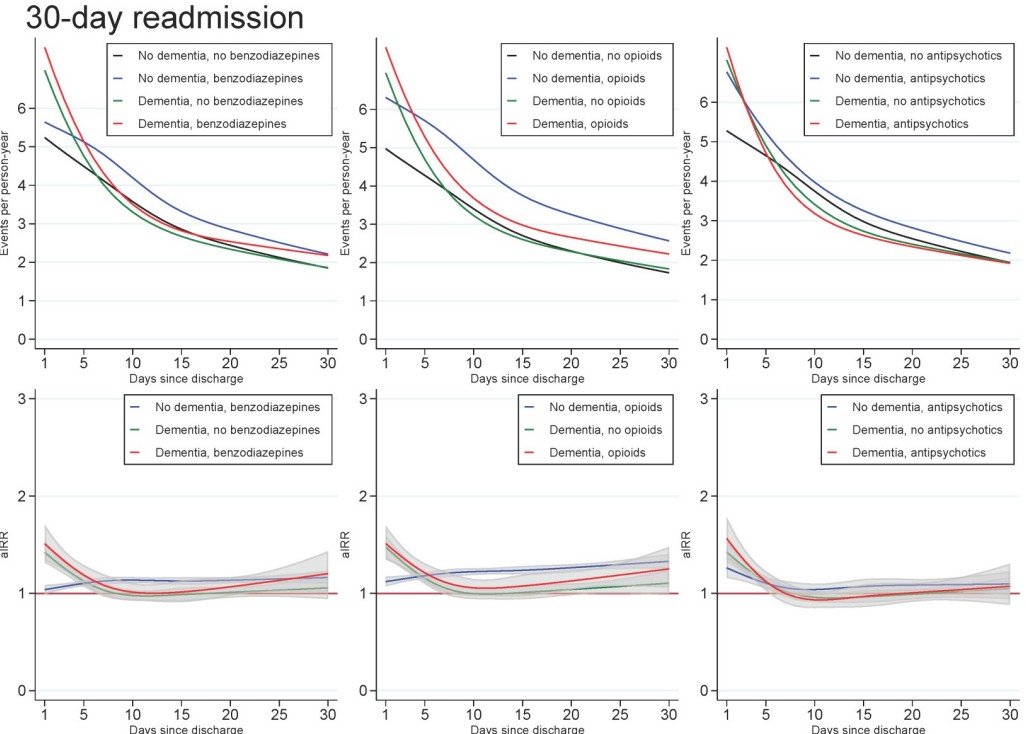

**Fig 4. Unadjusted readmission rates and adjusted incidence (readmission) rate ratios for patients with dementia (without medication use), patients with use of benzodiazepines, opioids or antipsychotics (without dementia), and patients with both or neither exposures, plotted as a function of days since discharge.** Abbreviations: aIRR: adjusted incidence (readmission) rate ratios. IRRs adjusted for sex, age, calendar period, cohabitation status, length of stay, type of pneumonia diagnosis, time since discharge, somatic comorbidities, psychiatric comorbidities, and alcohol/substance abuse.

have high validity; the positive predictive value of the pneumonia diagnosis is 90% [16] and that of dementia is 86% [49], and there is essentially complete registration of all deaths [15].

However, the study also had some limitations. First, we found a dementia prevalence of 8.7%, which is in concordance with another register-based study on dementia in hospitalized pneumonia patients [50], but lower than the prevalence of 18–42% reported in studies performing cognitive testing upon acute admissions [51, 52]. Yet, dementia should be assessed in a person who is not acutely unwell to avoid falsely misclassifying, for instance, delirium as dementia, and therefore, these studies may have overestimated the dementia prevalence. Second, as dementia is known to be underdiagnosed in general [53], we might not have included all cases of dementia in our exposure. However, this would imply that our estimates are conservative due to the containment of un-diagnosed persons with dementia in our reference group. Third, we defined use of benzodiazepines, opioids, and antipsychotics according to redeemed prescriptions. However, we cannot rule out secondary non-adherence (i.e. the prescription was redeemed but was not used accordingly [54]). Our intention was to investigate whether current use of these medications could serve as a marker of risk for readmission and mortality in persons with dementia. Therefore, we dichotomized individuals as either "non-users" or "users" of benzodiazepines, opioids, and antipsychotics without taking the intensity and the duration of the treatment into account. However, we cannot rule out that the associations could have varied with intensity and duration of treatment. Fourth, some individuals with dementia in our study might have been discharged due to (expected) impending death, which would contribute

to the higher short-term mortality found in our study. Yet, our data does not enable us to distinguish between expected and unexpected deaths. Fifth, due to the observational nature of the data, residual confounding cannot be excluded as the registers contain no information on in-hospital clinical measures, medications administered during hospitalizations, life style habits, and diagnoses registered only in primary care (such as alcohol abuse).

## Conclusions

This study explored the prognosis for patients with dementia versus patients without dementia in the transition from hospital to home and primary care management after discharge for pneumonia. Dementia was associated with a higher risk of 30-day mortality and readmission. The high mortality risk was markedly pronounced in those using antipsychotics, and the high readmission risk was primarily found within the first days after discharge.

Knowledge on the prognosis is needed to inform shared decision-making between the health-care professionals, patients and their caregivers, and to set goals of care for the post-discharge period. In the event of increased risks of readmission and mortality, care planning ought to take these higher risks into account.

## Supporting information

**S1 Appendix. Information on socioeconomic factors obtained from the Civil Registration System and Statistics Denmark.** [a]Highest level of education was classified according to the UNESCO classification as low (< 10 years), middle (10–15 years), and higher education (> 15 years). Due to unknown level of education in the earlier calendar period, this variable was limited to the last calendar period (i.e. 2012–2016).
(DOCX)

**S2 Appendix. Information on type of residency and home care services obtained from Statistics Denmark.** [a]Assessment of residency in nursing home in the month preceding index admission. Nursing homes are for permanent residents, and the nursing homes are staffed throughout the day by health care professionals. [b]Assessment of home care in the month preceding index admission. Home care is delivered to individuals living at home who are unable to manage everyday life on their own.
(DOCX)

**S3 Appendix. Information on comorbidity by the Multimorbidity Index obtained from the Danish National Patient Register, the Danish Psychiatric Central Register, and the Danish National Prescription Registry.** [a]Modified from original index by removing the categories "Thyroid disorder" and "Gout". [b]Modified from original index by removing the category"Allergy". [c]Modified from original index by adding the category "Dysphagia" (ICD-10: R13). [d]Modified from original index by removing the ATC code N02A. [e]Modified from original index by removing the category "HIV/AIDS" and renamed "Anemias". [f]Modified from original index by removing the categories "Vision problem", "Hearing problem", and "Migraine". [g]Modified from the original index by including epilepsy diagnoses only in the definition of the category "Epilepsy". [h]Modified from original index by removing the category "Anorexia/bulimia" and "Dementia" (which is the exposure in this study). [i]Modified from original index by removing N06AX12, which is used for smoking cessation.
(DOCX)

**S4 Appendix. Information on medication use obtained from the Danish National Prescription Registry.** [a]Remaining medication available in the data set, not already adjusted for in the

Comorbidity Index (S3 Appendix).
(DOCX)

**S5 Appendix. Information on index admission obtained from the Danish National Patient Register.** [a]If a pneumonia diagnosis was registered as both a primary and a secondary diagnosis during the same admission, it was categorized as a primary diagnosis.
(DOCX)

**S1 Table. Adjusted mortality rate ratios (aMRRs) for the risk of 30-day mortality in pneumonia patients with dementia (defined in five different ways) compared with those without in 298,872 admissions.** Abbreviations: aMRR: adjusted mortality rate ratio; CI: confidence interval. [a]All analyses were adjusted for sex, age, calendar period, cohabitation status, length of stay, type of pneumonia diagnosis, time since discharge, non-psychiatric comorbidities, psychiatric comorbidities, and alcohol/substance abuse.
(DOCX)

**S2 Table. Adjusted mortality rate ratios (aMRRs) for the risk of 30-day mortality in pneumonia patients with dementia (without medication use), with use of benzodiazepines, opioids or anti-psychotics (within the preceding four months) (without dementia), or with both dementia and medication use, compared with those with neither exposure in 298,872 admissions.** Abbreviations: aMRR: adjusted mortality rate ratio; CI: confidence interval, AP: attributable proportion. [a]Adjusted for sex, age, calendar period, cohabitation status, length of stay, type of pneumonia diagnosis, time since discharge, somatic comorbidities, psychiatric comorbidities, and alcohol/substance abuse. [b]Attributable proportion was calculated as: AP = (aMRR$_{(dementia+medication+)}$−aMRR$_{(dementia+ medication-)}$−aMRR$_{(dementia- medication+)}$ + 1) / aMRR$_{(dementia+ medication+)}$. [c]Excess number of events due to interaction was calculated as N$_{excess}$ = Number of events$_{(dementia+ medication+)}$ $^*$ AP.
(DOCX)

**S3 Table. Adjusted incidence rate ratios (aIRRs) for the risk of 30-day readmission in pneumonia patients with dementia (defined in five different ways) compared with those without in 298,872 admissions.** Abbreviations: aIRR: adjusted incidence rate ratio; CI: confidence interval. [a]All analyses were adjusted for sex, age, calendar period, cohabitation status, length of stay, type of pneumonia diagnosis, time since discharge, somatic comorbidities, psychiatric comorbidities, and alcohol/substance abuse.
(DOCX)

**S4 Table. Adjusted incidence rate ratios (IRRs) for the risk of 30-day readmission in pneumonia patients with dementia (without medication use), with use of benzodiazepines, opioids or anti-psychotics (within the preceding four months) (without dementia), or with both, compared with those with neither exposure in 298,872 admissions.** Abbreviations: aIRR: adjusted incidence rate ratio; CI: confidence interval, AP: attributable proportion. [a]Adjusted for sex, age, calendar period, cohabitation status, length of stay, type of pneumonia diagnosis, time since discharge, somatic comorbidities, psychiatric comorbidities, and alcohol/substance abuse. [b]Attributable proportion was calculated as: AP = (aIRR(dementia+ medication+)−aIRR(dementia+ medication-)−aIRR$_{(dementia- medication+)}$ + 1) / aIRR$_{(dementia+ medication +)}$. [c]Excess number of events due to interaction was calculated as N$_{excess}$ = Number of events$_{(dementia+ medication+)}$ $^*$ AP.
(DOCX)

**S5 Table. Adjusted mortality rate ratios (aMRRs) for the risk of 30-day mortality in pneumonia patients with dementia compared with those without in 2012–2016 in 106,949**

**admissions.** Abbreviations: aMRR: adjusted mortality rate ratio; CI: confidence interval.
[a]Main analysis was restricted to the years 2012–2016 and adjusted for sex, age, calendar period, cohabitation status, length of stay, type of pneumonia diagnosis, time since discharge, somatic comorbidities, psychiatric comorbidities, and alcohol/substance abuse. [b]Further adjusted for educational level, type of residency, and extent of home care.
(DOCX)

**S6 Table. Adjusted incidence rate ratios (aIRRs) for the risk of 30-day readmission in pneumonia patients with dementia compared with those without in 2012–2016 in 106,949 admissions.** Abbreviations: aIRR: adjusted incidence rate ratio; CI: confidence interval. [a]Main analysis was restricted to the years 2012–2016 and adjusted for sex, age, calendar period, cohabitation status, length of stay, type of pneumonia diagnosis, time since discharge, somatic comorbidities, psychiatric comorbidities, and alcohol/substance abuse. [b]Further adjusted for educational level, type of residency, and extent of home care.
(DOCX)

## Acknowledgments

All authors contributed to the design of the study, interpretation of the data, drafting and revising the paper and provided final approval of the version to be published. SBG had access to all data in the study and had final responsibility for the content of the manuscript, including all analyses. All authors agree to be accountable for all aspects of the work. The authors would like to thank Dr Matthew Lewis, Professor Andrea Maier, Professor Paul Komesaroff, and Associate Professor Rosie Watson for research meetings that contributed to this study. Further, the authors would like to thank the Danish Health Data Authority and Statistics Denmark for providing the data, and Lone Niedziella for language revision of the manuscript.

## Author Contributions

**Conceptualization:** Susanne Boel Graversen, Henrik Schou Pedersen, Annelli Sandbaek, Catherine Hauerslev Foss, Anette Riisgaard Ribe.

**Data curation:** Susanne Boel Graversen, Henrik Schou Pedersen.

**Formal analysis:** Susanne Boel Graversen, Henrik Schou Pedersen, Anette Riisgaard Ribe.

**Funding acquisition:** Susanne Boel Graversen.

**Investigation:** Susanne Boel Graversen, Henrik Schou Pedersen, Annelli Sandbaek.

**Methodology:** Susanne Boel Graversen, Henrik Schou Pedersen, Annelli Sandbaek, Victoria Jane Palmer, Anette Riisgaard Ribe.

**Project administration:** Susanne Boel Graversen, Annelli Sandbaek, Anette Riisgaard Ribe.

**Resources:** Susanne Boel Graversen, Victoria Jane Palmer.

**Supervision:** Henrik Schou Pedersen, Annelli Sandbaek, Catherine Hauerslev Foss, Anette Riisgaard Ribe.

**Writing – original draft:** Susanne Boel Graversen.

**Writing – review & editing:** Susanne Boel Graversen, Henrik Schou Pedersen, Annelli Sandbaek, Catherine Hauerslev Foss, Victoria Jane Palmer, Anette Riisgaard Ribe.

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
