## [Decision Letter · Decision Letter 0]

31 Dec 2020

PONE-D-20-37221

The significance of dementia and use of benzodiazepines, opioids, and antipsychotics for short-term readmission and mortality after a pneumonia admission

PLOS ONE

Dear Dr. Graversen,

Thank you for submitting your manuscript to PLOS ONE. After careful consideration, we feel that it has merit but does not fully meet PLOS ONE’s publication criteria as it currently stands. Therefore, we invite you to submit a revised version of the manuscript that addresses the points raised during the review process.

The manuscript by Graversen et al. was well assessed by the two Reviewers.

Minor revisions are necessary before acceptance. See the Reviewers' comments carefully and respond them appropriately.

We look forward to receiving your revised manuscript.

Kind regards,

Masaki Mogi

Academic Editor

PLOS ONE

Journal Requirements:

Reviewers' comments:

Reviewer's Responses to Questions

**Comments to the Author**

1. Is the manuscript technically sound, and do the data support the conclusions?

Reviewer #1: Yes

Reviewer #2: Yes

2. Has the statistical analysis been performed appropriately and rigorously? 

Reviewer #1: I Don't Know

Reviewer #2: Yes

3. Have the authors made all data underlying the findings in their manuscript fully available?

Reviewer #1: Yes

Reviewer #2: Yes

4. Is the manuscript presented in an intelligible fashion and written in standard English?

Reviewer #1: Yes

Reviewer #2: Yes

5. Review Comments to the Author

Reviewer #1: Thank you for the opportunity to read this interesting study. I have a few suggestions, below, but this paper needs to be reviewed by a statistician.

Abstract

This should contain the raw numbers for the main outcomes. Readers need clarification as to the covariates in the adjustment.

Introduction

Use of opioids to control behaviour is unusual, and the justification should be included.

Methods

What is meant by a ‘validated’ diagnosis?

Were people excluded if they were readmitted within a year? (lines 83,84).

If only a single prescription was issued for any medicine class, was this treated as if the medicine had been prescribed regularly? If only a single prescription was issued, ever, there is a possibility that the medicine was not used or used very little. If this is not accounted in the analysis, I suggest it is mentioned in the limitations.

The covariates are clarified in Table 1, and the text should point readers to this. A definition of a ‘nursing home’ in Denmark should be offered.

Alcohol problems and substance misuse are risk factors distinct from illness, and both can affect respiration. If these cannot be accounted separately, i suggest this is commented in the limitations.

p.7, line 123-4 gives a list of medicines as potential covariates (antibiotics etc). I did not see any results for these analyses. Is it possible that antibiotics were withheld from some patients? If so, did this affect outcome? These medicines should also appear in Table 1.

Results

Please could you explain a negative attributable proportion?

The figures are difficult to read without a colour printer.

Conclusions

This is an important and powerful analysis. These findings imply that, to reduce the risks of pneumonia, patients should be more closely monitored for the adverse effects of their prescription medicines, particularly antipsychotics, which sometimes cause hyper-salivation, impaired swallowing reflex, dyskinesia, and muscle rigidity [1-3].

Style

Line 88. If people had emigrated, I’m not sure how they appear in this study. Should this read ‘immigration’?

Line 296

Careful copy editing by a native speaker is needed. ‘Compared’ is followed by ‘with’, unless the comparison is with an abstract noun or concept.

Please could you send me a pdf on publication?

1. Jordan S, Banner T, Gabe-Walters M, Mikhail JM, Panes G, Round J, Snelgrove S., Storey S., Hughes D. (2019) Nurse-led medicines’ monitoring in care homes, implementing the Adverse Drug Reaction (ADRe) Profile improvement initiative for mental health medicines: An observational and interview study. PLoS ONE 14(9): e0220885. https://doi.org/10.1371/journal.pone.0220885

2. Jordan S, Gabe-Walters ME, Watkins A, Humphreys I, Newson L, Snelgrove S, Dennis M. (2015) Nurse-Led Medicines' Monitoring for Patients with Dementia in Care Homes: A Pragmatic Cohort Stepped Wedge Cluster Randomised Trial. PLoS ONE 10(10): e0140203. doi:10.1371/journal.pone.0140203

3. ADRE – THE ADVERSE DRUG REACTION PROFILE - HELPING TO MONITOR MEDICINES

Available: http://www.swansea.ac.uk/adre/

Reviewer #2: This is a register-based study investigating the readmission and mortality risk after pneumonia among people with dementia. Although the results are not terribly surprising, the topic is important, as pneumonia is a frequent and impactful outcome among people with dementia. The manuscript is compellingly written and the study is in my opinion well designed. I only have as few, fairly minor comments.

1) My greatest confusion about the manuscript is about the role of the antipsychotics, benzodiazepines and opioids in this study. The headline and the results section would have me believe that they are considered as an exposure, but they are quite simply analyzed and methods are briefly described in the section for covariates. The requirement for “use” is one filled prescription in the last six (or four) months, without any consideration for intensity or length of treatment. Would the authors consider this as a limitation to their study?

2) Related to the first point, I presume the authors did not analyze drug use during hospitalizations. This in my opinion should also be listed as a limitation to the study.

3) In the section on the study population, the authors mention that the included must have a “validated diagnosis of pneumonia”. Is this validation in reference to the study by Thomsen et al., or is it a validation of that specific pneumonia case? If the former, please state so in a separate sentence to avoid confusion; if the latter, please describe the validation process.

4) Could the authors kindly explain why the study population is restricted to people younger than 100 years?

5) In the instruction section, the use of sedative medication is referred to as a “dementia-related characteristic”. Kindly amend the sentence.

6. PLOS authors have the option to publish the peer review history of their article (what does this mean?). If published, this will include your full peer review and any attached files.

Reviewer #1: **Yes: **Sue Jordan

Reviewer #2: No

---

## [Author Response · Author response to Decision Letter 0]

10 Jan 2021

Reviewer #1:

Thank you for the opportunity to read this interesting study. I have a few suggestions, below, but this paper needs to be reviewed by a statistician.

Author’s response: 

Thank you for taking the time to review our work. We agree that the statistics should be done very careful when working with large datasets. This is why our second author is a statistician who has supervised all the analyses.

Abstract: 

1. This should contain the raw numbers for the main outcomes. Readers need clarification as to the covariates in the adjustment.

Author’s response: 

The raw numbers for the main outcomes and the covariates in the adjustment are now stated in the abstract (please see page 2, lines 32-34 (with hidden tracked changes) in the revised manuscript).

Introduction:

2. Use of opioids to control behaviour is unusual, and the justification should be included.

Author’s response: 

Thank you for this comment. Underlying pain, which can be difficult to communicate due to cognitive limitations, is often a cause of agitation in persons with dementia. As the reviewer points out, this was not clearly stated in the manuscript. This has been amended (please see page 4, lines 54-56 in the revised manuscript).

Methods

3. What is meant by a ‘validated’ diagnosis?

Author’s response:

The validation is in reference to the study by Thomsen et al, which found a positive predictive value of 90% for these ICD-10 diagnoses (this is stated on page 17, line 318). In order to avoid confusion, we have deleted the word “validated” from the methods section since the positive predictive value is mentioned later in the manuscript (please see page 5, line 87 in the revised manuscript).

4. Were people excluded if they were readmitted within a year? (lines 83,84).

Author’s response:

In order to study an older, geriatric population, we wanted to allow persons to re-enter the cohort instead of including only first-time admissions. However, we wanted to study incident pneumonia cases and therefore, we introduced a pneumonia-free quarantine of twelve months before re-entrance was allowed. Persons with a pneumonia admission within these twelve months were not excluded, as they were still allowed to re-enter the cohort when twelve consecutive months had passed without a pneumonia admission. We have changed the wording in the manuscript in the effort to clarify these decisions (please see page 5, lines 88-92 in the revised manuscript).

5. If only a single prescription was issued for any medicine class, was this treated as if the medicine had been prescribed regularly? If only a single prescription was issued, ever, there is a possibility that the medicine was not used or used very little. If this is not accounted in the analysis, I suggest it is mentioned in the limitations.

Author’s response:

We defined use of medication in different ways in the study depending on whether we wanted to identify 1) users at any point in time (i.e. anti-dementia drugs) or 2) current users (i.e. benzodiazepines, opioids, and antipsychotics). 

1) We defined use of anti-dementia drugs as ≥1 redeemed prescription at any given time prior to the index admission (page 6, lines 107-109). This approach was taken due to the chronic nature of the dementia disease: if a person had an anti-dementia drug prescribed at any point in time, we consider that person to have dementia (regardless of the continuing use of the anti-dementia drug). 

2) We defined use of benzodiazepines, opioids, and antipsychotics as ≥1 redeemed prescription within the six months preceding index admission (page 6, lines 115-118). In order to challenge this cut-off, we restricted this to four months preceding index admission and found results similar to our main analysis. However, we cannot rule out the existence of secondary non-adherence to these medication treatments (i.e. a person has redeemed the prescription but does not use it according to the prescription). We have added this important point to the limitations (please see page 18, lines 329-330 in the revised manuscript). 

6. The covariates are clarified in Table 1, and the text should point readers to this. A definition of a ‘nursing home’ in Denmark should be offered.

Author’s response:

Readers are now pointed to Table 1 in the section on covariates (please see page 6, line 114 in the revised manuscript).

A definition of ‘nursing home’ has been added to the section of Study design and setting (please see page 5, lines 79-81 in the revised manuscript).

7. Alcohol problems and substance misuse are risk factors distinct from illness, and both can affect respiration. If these cannot be accounted separately, i suggest this is commented in the limitations.

Author’s response:

We agree with the reviewer that alcohol and substance abuse are both conditions that can affect respiration and we therefore chose to include these conditions as adjusting variables in our main model. We adjusted for these conditions in the same step as psychiatric comorbidities due to the frequent correlation of these conditions [1]. Since we are able to account for alcohol problems and substance misuse separately, we provide the analysis in the attached Response to Reviewers.

The results show that adjusting for these conditions separately provides estimates identical to the current main model. This may be due to small numbers of persons with these diagnoses, as our data does not allow us to identify persons with alcohol problems and substance abuse who are only cared for by the GP. We have added this as a limitation (please see page 18, line 341 in the revised manuscript).

8. p.7, line 123-4 gives a list of medicines as potential covariates (antibiotics etc). I did not see any results for these analyses. Is it possible that antibiotics were withheld from some patients? If so, did this affect outcome? These medicines should also appear in Table 1.

Author’s response:

Thank you for pointing out that antibiotics, antithrombotic drugs, glucocorticoids, antineoplastic agents, and immunosuppressants were unfortunately omitted from Table 1. We have corrected this now (please see Table 1 in the revised manuscript). Yet, in the analyses, these covariates were included: These covariates were treated as potential confounders and adjusted for in model 4. These results showed that adjusting for the mentioned medications did not change the main estimate in model 3. We agree with the reviewer that for some patients, antibiotics will be withheld, probably due to (expected) impending death. We have addressed this in the existing section on limitations (please see page 18, lines 335-338).

Results:

9. Please could you explain a negative attributable proportion?

Author’s response:

For those doubly exposed to both dementia and medication (e.g. opioids), a negative attributable proportion entails that the effect of combining the two exposures is in fact lower than expected based on the summation of the independent effects of each exposure. We have added this explanation to the manuscript (please see page 15, lines 241-243 in the revised manuscript). 

10. The figures are difficult to read without a colour printer.

Author’s response:

The colour scheme is within the RGB (8 bit/channel) as requested for publication in Plos One, however, we apoligize for the inconvience it may have caused.

Conclusions:

11. This is an important and powerful analysis. These findings imply that, to reduce the risks of pneumonia, patients should be more closely monitored for the adverse effects of their prescription medicines, particularly antipsychotics, which sometimes cause hyper-salivation, impaired swallowing reflex, dyskinesia, and muscle rigidity. 

Author’s response: 

Thank you for these comments and for bringing our attention to the results of these studies on adverse drug reactions, which we have added to the discussion (please see page 16, lines 292-295 in the revised manuscript).

Style:

12. Line 88. If people had emigrated, I’m not sure how they appear in this study. Should this read ‘immigration’?

Author’s response:

We included data on emigration in order to take this into account in our analysis. If a person left the country during a 30-day follow-up, this person no longer contributed with at-risk time.

Line 296

13. Careful copy editing by a native speaker is needed. ‘Compared’ is followed by ‘with’, unless the comparison is with an abstract noun or concept.

Author’s response:

After copy editing by one of the native speaking co-authors, this is now amended (please see page 2, line 23 and page 11, line 170 in the revised manuscript for examples of these amendments).

14. Please could you send me a pdf on publication? 

Author’s response:

I would be happy to. Thank you for your interest in this work.

References

1. Hasin DS, Stinson FS, Ogburn E, Grant BF. Prevalence, correlates, disability, and comorbidity of DSM-IV alcohol abuse and dependence in the United States: Results from the national epidemiologic survey on alcohol and related conditions. Arch Gen Psychiatry. 2007 Jul;64(7):830–42. Available from: https://pubmed.ncbi.nlm.nih.gov/17606817/

Reviewer #2:

This is a register-based study investigating the readmission and mortality risk after pneumonia among people with dementia. Although the results are not terribly surprising, the topic is important, as pneumonia is a frequent and impactful outcome among people with dementia. The manuscript is compellingly written and the study is in my opinion well designed. I only have as few, fairly minor comments.

Author’s response:

Thank you for these comments.

1. My greatest confusion about the manuscript is about the role of the antipsychotics, benzodiazepines and opioids in this study. The headline and the results section would have me believe that they are considered as an exposure, but they are quite simply analyzed and methods are briefly described in the section for covariates. The requirement for “use” is one filled prescription in the last six (or four) months, without any consideration for intensity or length of treatment. Would the authors consider this as a limitation to their study?

Author’s response:

We agree with the reviewer that the title could lead to the assumption that dementia and medication (i.e. benzodiazepines, opioids, and antipsychotics) are treated as equal exposures in this study. We aimed to investigate how the association between dementia and our main outcomes varied with use of these medications in order to identify “red flags” (e.g. use of antipsychotics) which the clinicians should be aware of in these patients. We see how our wording in the title can cause confusion, and we have therefore changed the title of the study to “Dementia and the risk of short-term readmission and mortality after a pneumonia admission”. 

The medications are dichotomized as “non-users” and “users” in order to make the results more easily interpretable for the clinician: If a patient has dementia and uses antipsychotics at the time of discharge (regardless of intensity and length of treatment), that serves as a marker of risk in that patient and the clinician should take it into consideration when discharging the patient. However, we recognize that a long-term user of a medication may be different from a new user and that high dosage users may be different from low dosage users, and we have added this to the limitations of the study (please see page 18, lines 330-335 (with hidden tracked changes) in the revised manuscript).

2. Related to the first point, I presume the authors did not analyze drug use during hospitalizations. This in my opinion should also be listed as a limitation to the study.

Author’s response:

We agree that drug use during hospitalization could be an important factor to consider, yet, these data are not available in the Danish registries. Currently, residual confounding in relation to “in-hospital clinical measures” is mentioned as a limitation on page 18, line 339. We have added “medications administered during hospitalizations” to this sentence (please see page 18, line 340 in the revised manuscript).

3. In the section on the study population, the authors mention that the included must have a “validated diagnosis of pneumonia”. Is this validation in reference to the study by Thomsen et al., or is it a validation of that specific pneumonia case? If the former, please state so in a separate sentence to avoid confusion; if the latter, please describe the validation process.

Author’s response:

The validation is in reference to the study by Thomsen et al, which found a positive predictive value of 90% for these ICD-10 diagnoses (this is stated on page 17, line 318). In order to avoid confusion, we have deleted the word “validated” from the methods section since the positive predictive value is mentioned later in the manuscript (please see page 5, line 87 in the revised manuscript).

4) Could the authors kindly explain why the study population is restricted to people younger than 100 years?

Author’s response:

We have limited socioeconomic data on those aged 100+ years in the cohort and therefore, we chose to not include them. 

5) In the instruction section, the use of sedative medication is referred to as a “dementia-related characteristic”. Kindly amend the sentence.

Author’s response:

The sentence has been amended (please see page 4, lines 47-50 in the revised manuscript).

---

## [Decision Letter · Decision Letter 1]

15 Jan 2021

Dementia and the risk of short-term readmission and mortality after a pneumonia admission

PONE-D-20-37221R1

Dear Dr. Susanne Boel Graversen,

We’re pleased to inform you that your manuscript has been judged scientifically suitable for publication and will be formally accepted for publication once it meets all outstanding technical requirements.

Kind regards,

Masaki Mogi

Academic Editor

PLOS ONE

Additional Editor Comments (optional):

No further comment.

Reviewers' comments:

Reviewer's Responses to Questions

**Comments to the Author**

1. If the authors have adequately addressed your comments raised in a previous round of review and you feel that this manuscript is now acceptable for publication, you may indicate that here to bypass the “Comments to the Author” section, enter your conflict of interest statement in the “Confidential to Editor” section, and submit your "Accept" recommendation.

Reviewer #1: All comments have been addressed

Reviewer #2: All comments have been addressed

2. Is the manuscript technically sound, and do the data support the conclusions?

Reviewer #1: Yes

Reviewer #2: Yes

3. Has the statistical analysis been performed appropriately and rigorously? 

Reviewer #1: I Don't Know

Reviewer #2: Yes

4. Have the authors made all data underlying the findings in their manuscript fully available?

Reviewer #1: Yes

Reviewer #2: Yes

5. Is the manuscript presented in an intelligible fashion and written in standard English?

Reviewer #1: Yes

Reviewer #2: Yes

6. Review Comments to the Author

Reviewer #1: A very useful paper. Nurses should know these findings. I should like to cite this paper later this year.

Reviewer #2: The authors have done a good job answering to the issues pointed out by the reviewers. I have no further comments.

7. PLOS authors have the option to publish the peer review history of their article (what does this mean?). If published, this will include your full peer review and any attached files.

Reviewer #1: **Yes: **Sue Jordan

Reviewer #2: No

---

## [Editor Report · Acceptance letter]

19 Jan 2021

PONE-D-20-37221R1 

Dementia and the risk of short-term readmission and mortality after a pneumonia admission 

Dear Dr. Graversen:

I'm pleased to inform you that your manuscript has been deemed suitable for publication in PLOS ONE. Congratulations! Your manuscript is now with our production department. 

Kind regards, 

on behalf of

Dr. Masaki Mogi 

Academic Editor

PLOS ONE